# The illusion of information adequacy

**Hunter Gehlbach**[1]*, **Carly D. Robinson**[2], **Angus Fletcher**[3]*

**1** Johns Hopkins University, Baltimore, MD, United States of America, **2** Stanford University, Stanford, CA, United States of America, **3** The Ohio State University, Columbus, OH, United States of America

* gehlbach@jhu.edu (HG); fletcher.300@osu.edu (AF)

**Data Availability Statement:** All relevant data for this study are publicly available from the OSF repository (http://doi.org/10.17605/OSF.IO/Q2UFB).

**Funding:** Dr. Hunter Gehlbach received start-up funds from Johns Hopkins University School of

## Abstract

How individuals navigate perspectives and attitudes that diverge from their own affects an array of interpersonal outcomes from the health of marriages to the unfolding of international conflicts. The finesse with which people negotiate these differing perceptions depends critically upon their tacit assumptions—e.g., in the bias of naïve realism people assume that their subjective construal of a situation represents objective truth. The present study adds an important assumption to this list of biases: *the illusion of information adequacy*. Specifically, because individuals rarely pause to consider what information they may be missing, they assume that the cross-section of relevant information to which they are privy is sufficient to adequately understand the situation. Participants in our preregistered study (*N* = 1261) responded to a hypothetical scenario in which control participants received full information and treatment participants received approximately half of that same information. We found that treatment participants assumed that they possessed comparably adequate information and presumed that they were just as competent to make thoughtful decisions based on that information. Participants' decisions were heavily influenced by which cross-section of information they received. Finally, participants believed that most other people would make a similar decision to the one they made. We discuss the implications in the context of naïve realism and other biases that implicate how people navigate differences of perspective.

## Introduction

*You don't know what you don't know.*

–Socrates

*There are known knowns. These are things we know that we know. There are known unknowns. That is to say, there are things that we know we don't know. But there are also unknown unknowns. There are things we don't know we don't know.*

–Donald Rumsfeld

A core challenge in navigating the social world, is how to negotiate differences of perception, attitude, and understanding [1]. Whether the context entails romantic couples, bosses and

Education. The funders had no role in study design, data collection and analysis, decision to publish, or preparation of the manuscript.

**Competing interests:** The authors have declared that no competing interests exist.

employees, or heads of state brokering multi-lateral decisions, everyone experiences instances in which different actors maintain divergent perspectives despite exposure to the same information.

The tacit assumptions that people bring to situations powerfully influence the resolution of these differences of perspective as well as the ongoing relationships between the parties. For example, psychologists have shown evidence for a default belief that one's personal, subjective views represent objective understandings of reality [2]. This notion of naïve realism [3, 4] represents one of many biases that exemplifies how our baseline assumptions—in this case, the assumption that one sees objective reality—can wreak havoc on attempts to navigate different perspectives. As naïve realists, individuals assume that rational others will agree with their reactions, behaviors, and opinions. When others disagree, presumably these other people (a) have been exposed to different information, (b) are unwilling or unable to logically proceed from objective evidence to a reasonable conclusion, or (c) are biased [4].

Naïve realism studies often focus on the divergent construals of prominent conflicts such as the abortion debate [3], affirmative action [5], the Israeli-Palestinian relationship [6], and other well-known current events [7]. Thanks to our neural architecture, perceivers conflate their subjective interpretation of the situation with objective reality (e.g., presuming one's opinion to be a consensus fact). In short, our brain's default setting makes it hard to view others' construals as reasonable [8].

We propose that an equally important tributary feeding the river of misunderstanding is *the illusion of information adequacy*—people tacitly assume that they have adequate information to understand a situation and make decisions accordingly. Yet, individuals often have no way of knowing what they don't know. For less prominent interpersonal issues, many misunderstandings arise in contexts where information may be less accessible to each party. From Socrates to Rumsfeld, people often acknowledge that there is much that they do not know, including a meta-awareness of "unknown unknowns." We argue that another default setting—comparable to naïve realists' assumptions that they see objective reality—is that people fail to account for the unknown unknowns. Accordingly, they navigate their social worlds confidently assuming that they possess adequate information. They form opinions, reify values, and behave in ways that suggest that they have sufficient information to make rational decisions—often without pausing to wonder how much they do not know.

For example, many drivers have pulled up behind a first car at a stop sign only to get annoyed when that car fails to proceed when traffic lulls at the intersection. Drivers of these second cars may assume they possess ample information to justify honking. Yet, as soon as a mother pushing her stroller across the intersection emerges from beyond their field of vision, it becomes clear that they lacked crucial information which the first driver possessed. This example highlights the distinction with naïve realism. In the driving example, both parties construe the situation identically—nobody should drive over pedestrians pushing baby strollers. Rather, it is the second driver's tacit assumption that they have *"enough"* information that precipitates the misunderstanding.

While seemingly trivial, this everyday example epitomizes a pervasive phenomenon. People constantly judge others' actions and beliefs because they assume that they possess enough relevant information to make fair evaluations. As a result, perceivers misunderstand others' attitudes, opinions, and beliefs even though these perceivers might share the same perspective were they privy to the same information.

To demonstrate the illusion of adequate information, we conducted two preregistered experiments in which participants recommended whether to merge two schools or keep them separate—fully documented in our preprint (https://osf.io/preprints/psyarxiv/fdu3m). Given the similar results of both studies, this article focuses on the second, replication experiment

because of its larger, more diverse sample, enhancements to our measures, and improved research design.

Our primary goal was (a) to experimentally test whether perceivers assumed that they possessed adequate information—i.e., that they would view the cross-section of information to which they were privy as being sufficiently relevant, important, adequate in quantity, etc. In addition, we expected that participants would: (b) feel competent to make decisions, (c) be strongly influenced by the cross-section of information that they saw, and (d) assume that most others would make similar decisions to the one they made. We also anticipated that, (e) if participants were later exposed to the full array of information, they would continue to endorse their original position [9]. Results provide broad support for our specific hypotheses that, in the context of this experiment, participants maintain the illusion that they have adequate information, and this illusion impacts several downstream outcomes related to decision-making.

## Materials and methods

The procedures for both studies were approved by the Johns Hopkins University Human Subjects Committee, and all participants provided written informed consent prior to participation by clicking a button that indicated their consent. Data collection occurred from June 7th to 8th, 2023. All survey measures and the three articles about the potential merging of schools that formed our main experimental manipulation (see the *Intervention* tab) can be found in our codebook: https://osf.io/pgezc. The data and STATA code for the study are available from https://osf.io/k37dg/.

### Participants

We recruited an initial convenience sample of 1501 participants within the United States using the online platform, Prolific. We excluded those who failed an attention check question prior to entering the study ($n = 240$). Our final sample ($N = 1261$) had 252 participants in the control group, 503 in the pro-merge condition, and 506 in the pro-separate condition.

The majority male sample (59%) was predominantly White (71%); others identified as Asian/ Pacific Islander (9%), Black (11%), or Other (3%), and 8% identified as Hispanic. Participants' mean age was 39.8 years ($sd = 13.6$), with a median education level of 3 years of college. Politically, 651 participants identified as liberals, 254 as moderates, and 356 as conservatives.

### Procedures

Participants read an article, "Our School Water is Disappearing," that described a school located in a region where the local aquifer was drying up. Consequently, the school faced a decision to risk staying put (and hope for more rain in the future) or to merge with another school (entailing other risks). We assigned participants to one of three conditions. The control group's version of the article presented information about seven features of the situation: three arguments described benefits of merging, three identified benefits of remaining separate, and one was neutral. The *pro-merge* treatment participants' article presented the three arguments describing the benefits of merging and the neutral piece of information; the *pro-separate* participants' article presented three arguments about the benefits of remaining separate and the neutral information. After reading the article and responding to initial questions, we randomly sub-divided each treatment group in half to either: (a) respond to a set of survey questions (regarding their perceptions of information adequacy, decision-making competence, consensus, and other exploratory measures) or (b) to read a second article that exposed them to the

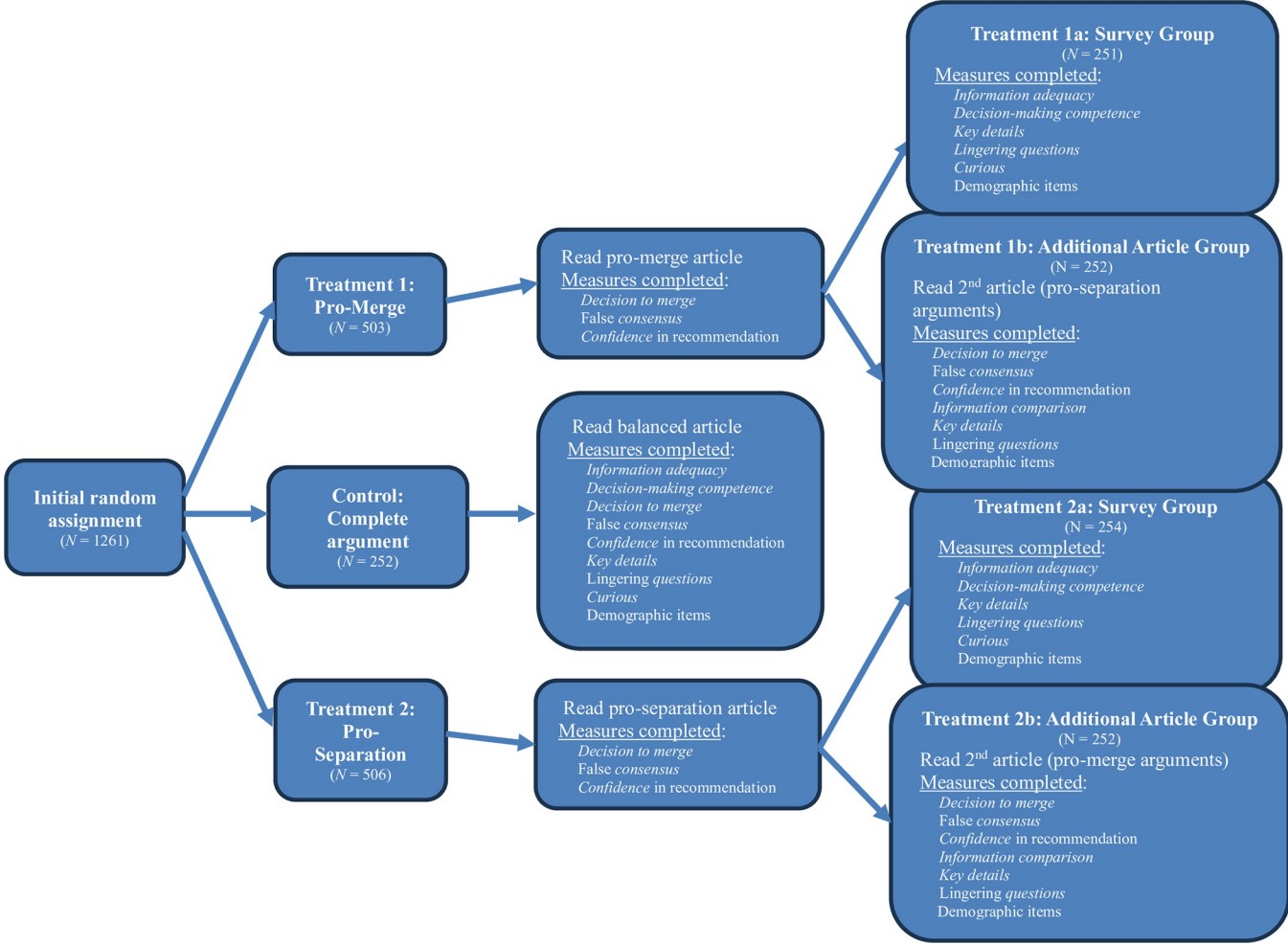

**Fig 1. Procedural flow for participants in different conditions.** The additional randomization within each treatment arm of the study allowed us to assess participants' beliefs about the information that they had (treatments 1a and 2a) and how those beliefs potentially changed after reading the second article (treatments 1b and 2b) without biasing respondents—i.e., had we asked participants how adequate their information was after the first article and then showed them a second article with countervailing information, they may have felt as though the experimenters were trying to trick them. The italicized words in the figure correspond to the names of the variables described in the measures section.

remaining arguments to merge or maintain separation of the two schools (i.e., providing them equivalent information to the control group) so that they could update their recommendations as they saw fit. See Fig 1 for details of the study design.

## Measures

Four distinct measures comprised the primary outcomes of interest in our five prespecified hypotheses. Our 5-item *information adequacy* scale ($a = .79$) assessed the extent to which participants felt as though the information they read in the article was sufficient in quantity, relevant, believable, important, and trustworthy enough to provide an "adequate" understanding of the situation the school was facing.

The 5-item *decision-making competence* scale ($a = .76$) measured students' perceptions that they could make a competent recommendation regarding the fate of the school based on the article that they read.

The *decision to merge* ($M = .55$) outcome was a binary choice. Participants responded to, "What decision would you recommend that the school board make." Response options were "Merge Prairie View into Landon" or "Keep the schools separate." For treatment groups 2a and 2b, this measure was asked a second time after these participants were exposed to the remaining arguments in a second article to test our 5th hypothesis.

Our *consensus* outcome was a single item ($M = .65$, $sd = .16$) designed to assess the false-consensus effect [10]. On a 0-to-100 slider, participants responded to, "About what percentage of people do you think would make the same decision as you did?"

To conduct exploratory analyses that could provide convergent data regarding our key outcome measures, several other measures were added. On 0–100 slider bar scales, all participants were asked to rate their *confidence* in their decision using a single item, "How confident are you that this is the smartest action for the school board to take?"

Treatment groups 1a, 2a, and the control group also responded to the following three items on 0–100 slider bars:

- "To what extent do you feel that you understand enough of the *key details* of the situation to make a good decision?"

- "To what extent do you feel as though you still have *questions* about important details of Prairie View's situation?"

- "How *curious* would you be to read a second article to learn more about this school's dilemma?"

For treatment participants who were instead routed to read a second article with the arguments they had not previously seen (i.e., groups 1b and 2b), we reassessed their recommendations using our *decision to merge*, *consensus*, and *confidence* measures. These participants then completed an *information comparison* scale ($a = .88$) which asked participants to compare the information between the first and second articles on criteria like relevance, trustworthiness, and importance. Finally, we reassessed these treatment participants' perceptions with the *key details* and *questions* items described above.

## Results

### Preliminary analyses

In line with our preregistration, we conducted a multinomial logistic regression to assess whether participant demographics predicted condition assignment. We found no evidence to that effect, $LR\chi^2(14) = 11.26$, $p = .665$, pseudo $R^2 = 0.005$. Thus, we did not employ any covariates in any of the models we report (i.e., there were no imbalances in condition assignment to correct for). We present the descriptive statistics and correlations for our primary variables of interest in Table 1.

### Pre-specified hypotheses

First, we wanted to assess whether participants perceived themselves as having adequate information about the situation even though the treatment groups received only half the information of the control group. Specifically, we tested whether the combined treatment groups 1a and 2a (who were routed directly to survey questions immediately after making their initial recommendations) would perceive the adequacy of the information that they received as comparable to the control group. Our OLS model supported this prespecified hypothesis $b_{Pooled} = 0.00$, 95% CI [-0.021, 0.022], adjusted $R^2 = -.001$. In addition, following Lakens et al.'s guidance [11], we conducted a two-sample *t*-test for mean equivalence ($a = .01$) to assess whether the

**Table 1. Descriptive statistics and correlations.**

| | | Mean | SD | N | 1 | 2 | 3 | 4 | 5 | 6 | 7 | 8 | 9 | 10 | 11 | 12 | 13 |
|---|---|---|---|---|---|---|---|---|---|---|---|---|---|---|---|---|---|
| 1) | Information adequacy | .69 | .14 | 1242 | – | | | | | | | | | | | | |
| 2) | Decision-making competence | .81 | .12 | 714 | .47*** | – | | | | | | | | | | | |
| 3) | Decision to merge | .55 | | 714 | .17*** | .04 | – | | | | | | | | | | |
| 4) | False consensus | .65 | .16 | 1183 | .34*** | .27*** | .16*** | – | | | | | | | | | |
| 5) | Confidence in recommendation | .70 | .21 | 1183 | .41*** | .33*** | .15*** | .56*** | – | | | | | | | | |
| 6) | Key details | .64 | .23 | 714 | .64*** | .37*** | .11** | .44*** | .54*** | – | | | | | | | |
| 7) | Lingering questions | .49 | .30 | 714 | .35*** | .17*** | .07 | .19*** | .25*** | .38*** | – | | | | | | |
| 8) | Curiosity | .70 | .27 | 714 | .14*** | .17*** | -.04 | .05 | .02 | .02 | -.02 | – | | | | | |
| 9) | Post: Decision to merge | .52 | | 466 | | | .31*** | .04 | -.00 | | | | – | | | | |
| 10) | Post: False consensus | .62 | .17 | 460 | | | .05 | .49*** | .22*** | | | | .07 | – | | | |
| 11) | Post: Confidence in recommendation | .66 | .21 | 460 | | | .02 | .31*** | .59*** | | | | .08 | .56*** | – | | |
| 12) | Post: Information comparison | .82 | .24 | 459 | | | -.03 | -.12* | -.17*** | | | | -.02 | .02 | -.07 | – | |
| 13) | Post: Key Details | .68 | .22 | 459 | | | .03 | .34*** | .45*** | | | | .06 | .41*** | .58*** | .01 | – |
| 14) | Post: Lingering questions | .51 | .30 | 459 | | | .02 | .14** | .15** | | | | .04 | .21*** | .25*** | -.04 | .30*** |

Note

* $p < .05$

**$p < .01$

***$p < .001$.

pooled treatment group and the control groups were statistically equivalent. The test revealed that the null effect was statistically indistinguishable from zero. Descriptively, the raw mean responses were closer to the "quite" (relevant, believable, trustworthy, etc.) than the "somewhat" response option for all three groups.

Next, we tested whether the pooled treatment groups 1a and 2a would perceive their ability to make a competent decision to be as high as the control group's perceived ability. Our OLS model predicting participants' scores on our *decision-making competence* scale supported this prespecified hypothesis also $b_{Pooled} = 0.00$, 95% CI [-0.018, 0.019], adjusted $R^2 = -.001$. The same two-sample equivalence test showed that we could reject the null hypothesis that the treatment effect is different than zero ($a = .01$); the pooled treatment and the control groups were statistically equivalent in their perceived decision-making competence [11]. In addition to the consistency of the means between the groups ($M_{Control} = .81$; $M_{Treatment\ Pro-Merge} = .81$; and $M_{Treatment\ Pro-Separate} = .80$), it is worth noting the relatively high values of the means—on average, participants responses were modestly higher than qualitative values corresponding to "quite" objective, careful, fair, etc. in evaluating the information and making a decision.

For our third prespecified hypothesis, we fit an OLS regression model to test for differences in the initial recommended decisions between our experimental conditions. Congruent with our hypothesis, we found that "pro-merge" treatment participants (groups 1a and 1b) were significantly more likely to recommend that the schools merge $b_{Pro-merge} = 0.33$, 95% CI [0.27, 0.40]; and the "pro-separation" participants (treatment groups 2a and 2b) were significantly more likely to recommend that the schools remain separate $b_{Pro-separate} = -0.32$, 95% CI [-0.38, -0.26], as compared to the control group adjusted $R^2 = .34$. See Fig 2.

Then we examined whether each group would presume that a majority of people would make the same recommendation that they had made—i.e., display the false consensus effect. To evaluate this prespecified hypothesis, we conducted three single-sample *t*-tests (for the control group, the pooled treatment group of 1a and 1b, and the pooled treatment group of 2a and 2b) to see whether each mean was significantly higher than 50%. We found that participants in

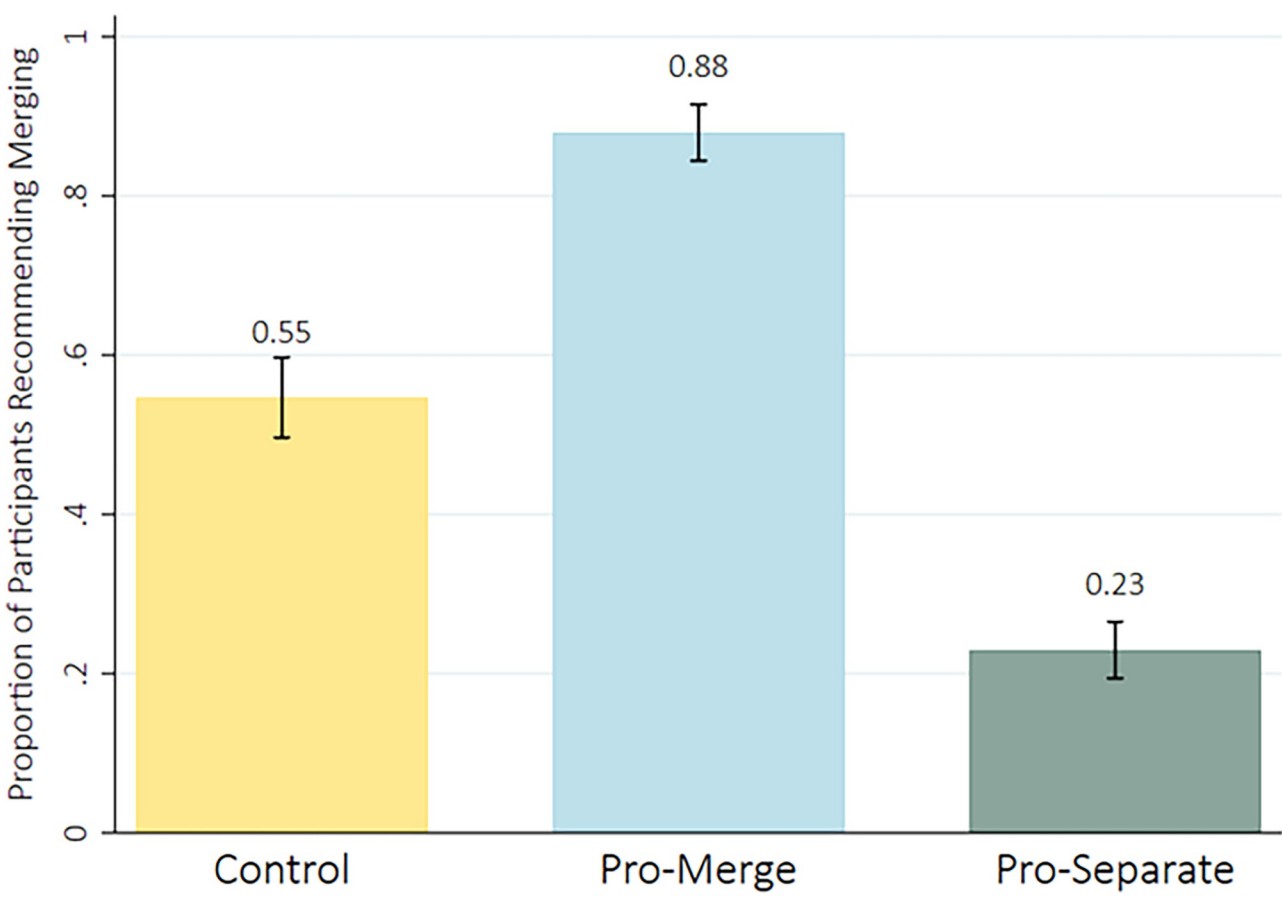

**Fig 2. Mean proportions of participants from each condition recommending merging and 95% CIs.**

each condition believed that the majority of other people would reach the same recommendation that they did; $M_{control} = .62$, $t_{(241)} = 5200$, 95% CI [.60, .63]; $M_{pro-merge} = .68$, $t_{(477)} = 6500$, 95% CI [.66, .69]; and $M_{pro-separate} = .65$, $t_{(462)} = 6800$, 95% CI [.63, .60].

Finally, we predicted that treatment groups 1b and 2b (who, after making their initial recommendations, read a second article providing the other half of the information that the control group received) would endorse their initial recommendation in significantly higher proportions than the control group (55% of whom recommended merging). In other words, we anticipated that the original information these treatment groups received—despite its partial nature—would help participants form opinions that would be hard to reverse, even in the face of learning compelling information to the contrary. Our data did not support this hypothesis. The majority of the participants in these two conditions did adhere to their original recommendation ($M_{Pro-merge} = .64$, $t_{(235)} = 1600$, 95% CI [.57, .70]; $M_{Pro-separate} = .68$, $t_{(229)} = 1600$, 95% CI [.62, .74]) after learning the additional information from the second article. However, overall, these two groups endorsed the recommendation to merge the two schools at comparable rates to the control group. See Fig 3.

## Exploratory findings

To further our understanding of the illusion of adequate information and the robustness of our prespecified findings, we included a series of additional measures for exploratory

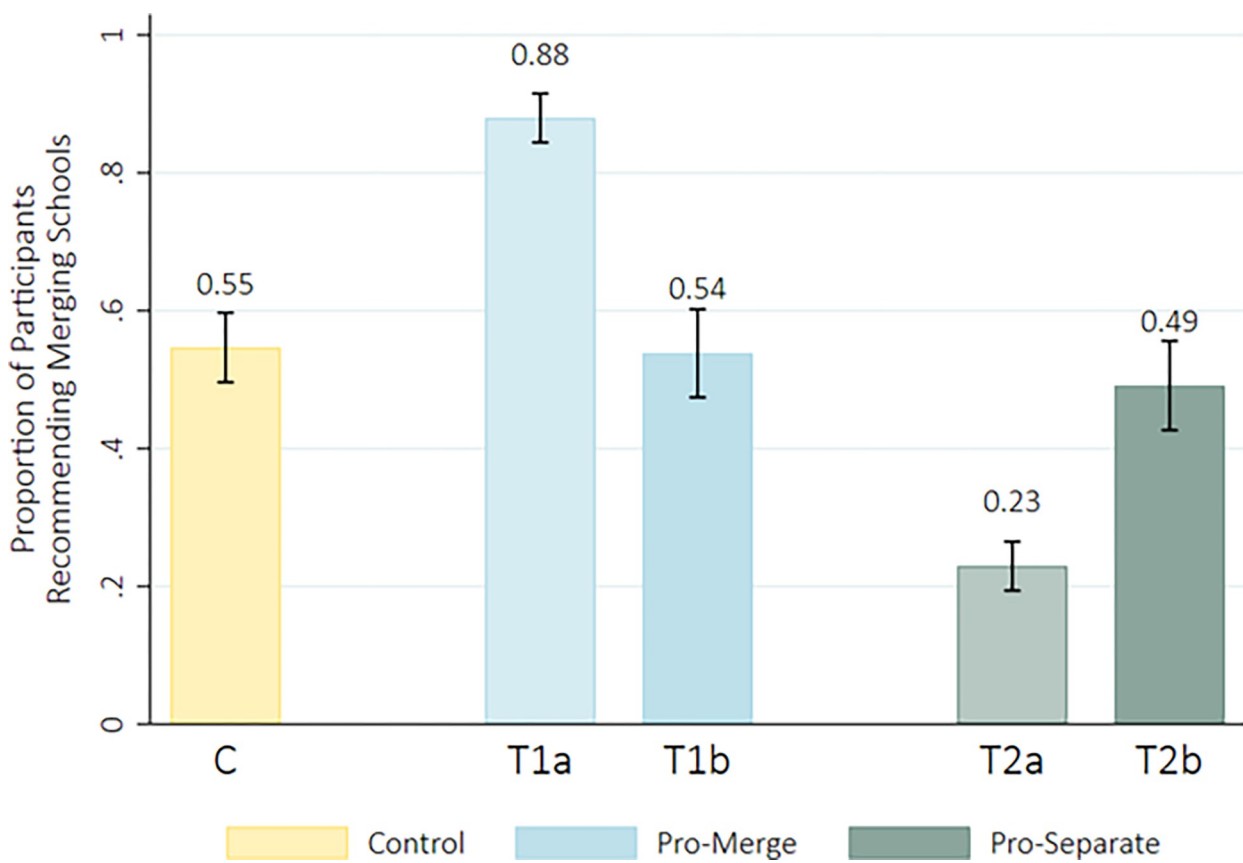

**Fig 3. Treatment groups' mean endorsements of merging recommendation before (T1) and after (T2) reading additional information with 95% CIs.**

purposes. As one example, each group also reported how confident they were about the recommendation that they provided. If people frequently fail to consider what they do not know about a situation, we anticipated that participants' confidence about their decisions would be largely unaffected by the amount of information they possessed.

As shown in S1 Fig, despite having half as much information, treatment groups 1a, 1b, 2a, and 2b all report greater mean levels of confidence in their initial recommendations than the control group. Pooling the means of all the treatment groups ($M_{PooledTreatment} = .71$, $sd = .21$) reveals that treatment participants actually reported significantly greater initial confidence than their control counterparts ($M_{Control} = .65$, $sd = .21$; $t_{(1181)} = 4.22$, $p < .001$, $d = -0.30$, 95% CI [-.45, -.16]). Participants in treatment groups 1b and 2b re-rated their confidence after exposure to the additional information, allowing for a within-subjects test of their confidence. A paired-sample $t$-test revealed that these participants became significantly less confident from their initial ratings ($M_{InitialPooled} = .71$, $sd = .22$) to their final ratings ($M_{FinalPooled} = .67$, $sd = .21$; $t_{(459)} = 5.031$, $p < .001$, $d = 0.17$, 95% CI [.06, .28]). In other words, across two different analyses we found that those with less information manifested greater confidence in their recommendations.

A second set of exploratory analyses further assessed how participants' perceptions of whether they had adequate information to make a decision changed after exposure to new information. S2 Fig shows participants' mean ratings on the 0–100 visual analog scales across all five branches of the study regarding whether they understood enough *key details* to make a

decision (Panel A) and whether they still had *questions* about the situation (Panel B). The means for each group hovered around two-thirds and half, respectively. These results provide additional, descriptive support for our core hypothesis that individuals assume they have adequate information to make recommendations even when some of them (the control group and treatment groups 1b and 2b) received twice as much information as others (treatment groups 1a and 2a) and despite all groups acknowledging that they still have some lingering questions.

## Discussion

Anecdotally, friction between parties holding divergent perceptions seems particularly prevalent in today's world—clashes over vaccines, abortion rights, who holds legitimate claims to which lands, and climate change surface in the news regularly. Conflicts on social media, between passengers and flight attendants, and among parents and referees at children's sporting events seem equally common. Correspondingly, the need to ascertain the various causes of poor perspective taking feels particularly acute at present.

Naïve realism offers one clear explanation of how our default assumptions can derail our attempts to understand others' perspectives [3]. We complement this line of research by proposing that a second major contributor to misunderstanding the perspectives of others is the illusion of adequate information.

Extending previous results (https://osf.io/preprints/psyarxiv/fdu3m), this study provides convergent evidence that people presume that they possess adequate information—even when they lack half the relevant information or be missing an important point of view. Furthermore, they assume a moderately high level of competence to make a fair, careful evaluation of the information in reaching their decisions. In turn, their specific cross-section of information strongly influences their recommendations. Finally, congruent with the second tenet of naïve realism [4] and the false consensus effect [10], our participants assumed that most other people would reach the same recommendation that they did.

Contrary to our expectations, although most of the treatment participants who ultimately read the second article and received the full array of information did stick to their original recommendation, the overall final recommendations from those groups became indistinguishable from the control group. Unlike Lord et al. [9], this result supports the idea that sharing or pooling of information may lead to agreement [4]. We suspect that prior attitudes regarding the topic in question may account for the contrasting results here. Lord et al. [9] focused on capital punishment—a well-known issue for which many hold strong prior opinions. We asked participants about a local, fictitious scenario—for which they were unlikely to hold strong prior opinions given the hypothetical nature of the scenario. Thus, changing the minds of our participants with new information was likely much easier than for participants in the Lord et al. [9] study.

We interpret our results as complementing the theory of naïve realism by illuminating an important, additional psychological bias that fuels people's assumption that they perceive objective reality and potentially undermines their motivation to understand others' perspectives [12]. Specifically, because people assume they have adequate information, they enter judgment and decision-making processes with less humility and more confidence than they might if they were worrying whether they knew the whole story or not.

How might greater doubt or humility facilitate the understanding and appreciation of others' perspectives? We suspect that in many real-world situations individuals rarely pause to consider how complete their picture of a situation is before passing judgment. Comedian George Carlin's observation—that, relative to our driving, slower drivers are idiots and faster drivers are maniacs—is often invoked in discussions of naïve realism, [e.g., 13]. Perhaps the

mere act of wondering, "do I have enough information?" could infuse humility and a willingness to allow more benefit of the doubt. Was the driver who just zoomed past en route to the hospital with a family emergency? Is the painfully slow driver transporting a fragile family heirloom? These types of questions might dissolve road rage (George Carlin's or our own) before it begins.

The illusion of adequate information may inform other theories as well. People may commit the fundamental attribution error more to the extent that they assume their knowledge about the situation is adequate [14]. Perhaps part of the reason stereotyping is a common approach to perceiving others [15] is because well-developed stereotypes allow for the illusion that perceivers have sufficient information about a target person. When predicting their own future emotional states, individuals may perform poorly because they assume that they have adequate information about their future circumstances; in reality, researchers have shown that people often fail to account for key details in their affective forecasting efforts [16, 17].

In addition to these theoretical implications, our findings may suggest practical strategies for improving people's social perspective taking capacities. Teaching individuals to pause and question how much information they know that they know about a situation—and, importantly, how much they might not know—could be a helpful disposition to cultivate. A concrete version of this strategy might entail listing relevant, lingering questions about a situation prior to passing judgment. Interventions in which people strategize how to learn more information about a situation or another party's beliefs might also be worthy of investigation.

Some readers may worry that our results seem so obvious as to be trivial. Our treatment participants had no way of knowing that they were deprived of a whole slate of arguments; naturally they would assume that they had adequate information. Others may worry that we stacked the deck by presenting the pro-merge participants with almost exclusively pro-merge arguments (and vice-versa for pro-separate participants). This concern, as well as the hypothetical scenario that may have seemed unimportant to our online participants, represent important limitations. At the same time, we suspect these features of our experiment represent exactly how this phenomenon unfolds in many real-world situations. People frequently have no way of knowing the extent to which the information they hold is complete or missing key elements. Relatedly, given polarized political and social media eco-systems, individuals are also regularly exposed to extremely unrepresentative cross-sections of information. Given strong motivations for cognitive efficiency [12, 18], people may not naturally want to expend extra effort considering what may not be known or how representative a sample of information is. Thus, our manipulation may serve as a reasonably prototypic illustration of how this bias unfolds in real world settings.

To be sure, this bias warrants more investigation. Future research that can investigate the generalizability of this phenomenon across a range of issues—including topics where people have prior knowledge and beliefs—is an important first step. We conceptualized "adequate" information broadly—asking participants to evaluate relevance, quantity, importance, trustworthiness, and credibility. Other studies that define the construct more narrowly—perhaps examining only the quantity of information provided—would provide additional insights into this phenomenon. Assuming similar evidence is found across issues and in real-world settings, then testing interventions to mitigate this bias and its downstream effects, will be another important contribution to this research agenda.

Although people may not know what they do not know, perhaps there is wisdom in assuming that some relevant information is missing. In a world of prodigious polarization and dubious information, this humility—and corresponding curiosity about what information is lacking—may help us better take the perspective of others before we pass judgment on them.

## Supporting information

**S1 Fig. Initial confidence in decision by condition and 95% CIs.** Comparison of the four different treatment groups versus the control group in their initial responses to the question, "How confident are you that your recommendation is the smartest action for the school board to take?".
(TIF)

**S2 Fig. Panel A. Perceived Key Details Received by Condition and 95% CIs.** Comparison of the four different treatment groups versus the control group in their responses to the question, "Sometimes people are satisfied that they have all the information that they need to make a decision. Other times, they feel that they need more information. After reading the article, to what extent do you feel that you understand enough of the key details of the situation to make a good decision?", **Panel B. Perceived Lack of Lingering Question by Condition and 95% CIs.** Comparison of the four different treatment groups versus the control group in their responses to the question, "To what extent do you feel as though you still have questions about important details of Prairie View's situation?".
(ZIP)

## Acknowledgments

The authors are grateful for the helpful thoughts and feedback provided by Andrew Ward.

## Author Contributions

**Conceptualization:** Hunter Gehlbach, Carly D. Robinson, Angus Fletcher.

**Data curation:** Carly D. Robinson.

**Formal analysis:** Carly D. Robinson.

**Investigation:** Hunter Gehlbach, Carly D. Robinson, Angus Fletcher.

**Methodology:** Hunter Gehlbach, Carly D. Robinson, Angus Fletcher.

**Project administration:** Hunter Gehlbach.

**Resources:** Hunter Gehlbach.

**Supervision:** Hunter Gehlbach.

**Writing – original draft:** Hunter Gehlbach, Carly D. Robinson, Angus Fletcher.

**Writing – review & editing:** Hunter Gehlbach, Carly D. Robinson, Angus Fletcher.

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
