## [Decision Letter · Decision Letter 0]

17 Jun 2024

PONE-D-24-13838

The illusion of information adequacy

PLOS ONE

Dear Dr. Gehlbach,

Thank you for submitting your manuscript to PLOS ONE. After careful consideration, we feel that it has merit but does not fully meet PLOS ONE’s publication criteria as it currently stands. Therefore, we invite you to submit a revised version of the manuscript that addresses the points raised during the review process.

Dear Author

Greetings

Thank you for considering publishing in our journal. I read your article and found it interesting and valuable. It was transferred to an evaluation process by a reviewer.

After receiving a detailed response from the reviewer, I recommending that you incorporate adjustments and responses in the revised version of the article.

I recommend paying special attention to the comments about each part of the article, and attaching a reference to each comment with an explanation of how it was taken into account when submitting the revised version.

Best regards

Gal Harpaz PhD.

We look forward to receiving your revised manuscript.

Kind regards,

Gal Harpaz, Ph.D.

Academic Editor

PLOS ONE

Journal Requirements:

Reviewers' comments:

Reviewer's Responses to Questions

**Comments to the Author**

1. Is the manuscript technically sound, and do the data support the conclusions?

Reviewer #1: Yes

2. Has the statistical analysis been performed appropriately and rigorously? 

Reviewer #1: Yes

3. Have the authors made all data underlying the findings in their manuscript fully available?

Reviewer #1: No

4. Is the manuscript presented in an intelligible fashion and written in standard English?

Reviewer #1: Yes

5. Review Comments to the Author

Reviewer #1: The manuscript reports on a preregistered online survey study using data gained with Prolific. The authors investigated the psychological basis of the epistemic phenomenon of naive realism, in particular how a lack of information affects judgments in light of people's general impression of having enough information at their disposal. The authors varied the amount of information their participants received, with those receiving less information being confident of knowing enough about the issue. The data support the claim that people imagine others to arrive at similar conclusions while they were in fact strongly influenced in their decision by the selection of information they were presented with.

The study offers an insight into decision-making under limited information availability and resulting biases. I have multiple issues with the manuscript:

- The authors claim that their study revolves around a "local, hypothetical scenario—for which they could not have held previous beliefs." I do not share the assumption that participants read the scenario without any preexisting biases. Any merger is associated with several consequences, such as lay-offs, but also cost reductions. Nevertheless, this limitation apparently has not affected judgments substantially, but should still be discussed.

- In the case of equivalence t-tests, the authors should mention the exact name of the test they used, since multiple implementations are available.

- It is unclear from the manuscript why some treatment groups completed different measures than others. This is rather confusingly presented in Figure 1 and the authors do not explain why the names of some measures are italicized.

- Although the PLOS ONE data sharing guidelines specify that data must be shared with reviewers unless a reason is given, the authors only plan on making the data available after acceptance without any justification. This violates the requirements of the journal and would need to be fulfilled.

The Rumsfeld quote is very illustrative.

6. PLOS authors have the option to publish the peer review history of their article (what does this mean?). If published, this will include your full peer review and any attached files.

Reviewer #1: No

---

## [Author Response · Author response to Decision Letter 0]

15 Jul 2024

Please see the appended "Response to Reviewers" letter.

---

## [Editor Report · Decision Letter 1]

22 Jul 2024

PONE-D-24-13838R1The illusion of information adequacyPLOS ONE

Dear Dr. Gehlbach,

Thank you for submitting your manuscript to PLOS ONE. After careful consideration, we feel that it has merit but does not fully meet PLOS ONE’s publication criteria as it currently stands. Therefore, we invite you to submit a revised version of the manuscript that addresses the points raised during the review process.

Dear Author

Thank you for resubmitting the article, we have received the additional reviewer's comments back, and now we have some final issues that require your attention.

Please pay attention to the four comments, we will wait for a revised version and ask that you address each of the issues the reviewer commented on.

We look forward to receiving your revised manuscript.

Kind regards,

Gal Harpaz, Ph.D.

Academic Editor

PLOS ONE
---

## [Author Response · Author response to Decision Letter 1]

15 Aug 2024

I have changed the titles of the three documents inquired about to exactly match the request in the email. Please note that the email asks, "Please pay attention to the four comments" but then only lists three bulleted items (requesting changes to the file names that were uploaded previously). If there is a fourth issue we need to address, please let us know.

Thanks,

Hunter

7/25/24

Please note my responses below to the latest email you sent on 7/24/24...

1. We note that the grant information you provided in the ‘Funding Information’ and ‘Financial Disclosure’ sections do not match.

When you resubmit, please ensure that you provide the correct grant numbers for the awards you received for your study in your cover letter; we will change the online submission form on your behalf.

I have noted in the response to reviewers' letter that I have done the best I can in the submission system, have been as accurate as possible, and need your specific guidance on how to proceed if you want something different.

2. Please note that your Data Availability Statement is currently missing the repository name. If your manuscript is accepted for publication, you will be asked to provide these details on a very short timeline. We therefore suggest that you provide this information now, though we will not hold up the peer review process if you are unable.

I'm not sure what is being asked for here. In our data availability statement in the submission system we state: "Our data and code are posted at: https://osf.io/k37dg/". Our response to reviewers contains this URL too. In the manuscript, you will see this on p. 6. I have added a DOI number and that the name of the hosting site is "Open Science Framework" just in case that was what you wanted?

3. Please change the highlighted text in your manuscript with black text.

I have looked through the submitted manuscript and do not see any highlighted text. I have changed all font to black in case that was what you wanted?

Please note that I am happy to make whatever changes you want but need more specificity. If there are any final requests from your end please list all of them--I know your time is as valuable as mine is.

Thank you,

Hunter

---

## [Editor Report · Decision Letter 2]

27 Aug 2024

The illusion of information adequacy

PONE-D-24-13838R2

Dear Dr. Gehlbach,

We’re pleased to inform you that your manuscript has been judged scientifically suitable for publication and will be formally accepted for publication once it meets all outstanding technical requirements.

Kind regards,

Gal Harpaz, Ph.D.

Academic Editor

PLOS ONE

Additional Editor Comments (optional):

Dear Authors,

Thank you for resubmitting the article. It is interesting, important and very relevant for the present time. Regarding editorial matters, if necessary they contacted you on the matter.

Best regards
---

## [Editor Report · Acceptance letter]

13 Sep 2024

PONE-D-24-13838R2 

PLOS ONE

Dear Dr. Gehlbach, 

I'm pleased to inform you that your manuscript has been deemed suitable for publication in PLOS ONE. Congratulations! Your manuscript is now being handed over to our production team.

Kind regards, 

on behalf of

Dr. Gal Harpaz 

Academic Editor

PLOS ONE